# Associations Between Body Image, Eating Behaviors, and Diet Quality Among Young Women in New Zealand: The Role of Social Media

**DOI:** 10.3390/nu16203517

**Published:** 2024-10-17

**Authors:** Jessica A. Malloy, Hugo Kazenbroot-Phillips, Rajshri Roy

**Affiliations:** 1Discipline of Nutrition and Dietetics, School of Medical Sciences, Faculty of Medical and Health Sciences, University of Auckland, 85 Park Road Grafton, Auckland 1011, New Zealand; jmal232@aucklanduni.ac.nz (J.A.M.);; 2Discipline of Nutrition and Dietetics, Susan Wakil School of Nursing and Midwifery, Faculty of Medicine and Health, The University of Sydney, Camperdown, NSW 2050, Australia; 3Charles Perkins Centre, The University of Sydney, Camperdown, NSW 2050, Australia

**Keywords:** body image, diet quality, eating behaviors, young women, social media

## Abstract

**Background/Objectives**: This study investigates the relationship between diet quality and body image disturbance among young women aged 18–24, a crucial period for establishing lifelong health behaviors. Given the increasing exposure to social media, which often promotes unrealistic beauty standards, this research aims to explore associations between eating behaviors, diet quality, and body image disturbance. **Methods**: A mixed-methods approach was employed, combining qualitative focus group discussions with quantitative analysis. Focus groups (*n* = 19) explored themes of body image dissatisfaction. The Body Image Disturbance Questionnaire (BIDQ) was administered to 50 participants (young women aged 18–24) to quantitatively assess body image disturbance, while diet quality was evaluated using the Australian Recommended Food Scores (ARFS). The Three-Factor Eating Questionnaire (TFEQ-R18) was also used to assess eating behaviors, including cognitive restraint, uncontrolled eating, and emotional eating. A social influence questionnaire (SIQ) was administered to measure the effect of social influence. Pearson’s correlation coefficient was used to determine the relationship between ARFS, BIDQ, and TFEQ-R18 scores. **Results**: Qualitative findings revealed persistent dissatisfaction with body shape, largely influenced by social media. Quantitatively, 65% of participants scored above the clinical threshold for body image disturbance (mean BIDQ score = 4.2, SD = 0.8). The correlation between ARFS and BIDQ scores was weak and not statistically significant (r = 0.057, *p* = 0.711). However, a significant positive correlation was observed between time spent on social media and body image disturbance (r = 0.58, *p* < 0.01). Additionally, TFEQ-R18 results indicated that 45% of participants displayed moderate levels of uncontrolled eating, and 36.5% demonstrated moderate levels of emotional eating. **Conclusions**: While social media is associated with body image concerns, its effect on eating behaviors and diet quality shows weak correlations, suggesting that other factors may mediate these outcomes. These results suggest the complexity of the associations between body image, eating behaviors, and diet quality, indicating that interventions should consider psychological drivers behind these concerns alongside social media usage.

## 1. Introduction

Body image disturbance is a significant psychological concern among young women, particularly those aged 18–24, who are at a crucial stage for establishing long-term health behaviors [1]. Body image refers to an individual’s perceptions, thoughts, and feelings about their physical appearance, which can significantly influence their self-esteem and overall mental health [2]. The pervasive presence of social media has been associated with shaping negative body perceptions [3]. This phenomenon has led to increased dissatisfaction with body shape and size, resulting in heightened awareness of perceived body imperfections and a greater risk of developing body image disturbances [3,4,5,6,7,8,9,10].

Studies suggest that body image is closely linked to eating behavior and diet quality. A quality diet is often characterized by its variety and balance, reflecting a wide range of nutrients essential for maintaining health and preventing chronic diseases [11]. Positive body image and autonomous exercise motivations have been linked to healthier eating outcomes, while body dissatisfaction and restrained eating are associated with poorer diet quality and lower dietary diversity [12,13,14,15,16,17,18,19,20,21]. While some studies suggest that poor diet quality may exacerbate body dissatisfaction [22], others indicate that the connection may be more complex, with factors such as social media exposure playing a more significant role [10].

Despite the increasing recognition of social media as a significant factor in shaping body image perception across various age groups and settings, gaps remain in understanding the nuanced relationships between social media, body image disturbance, and eating behaviors in diverse cultural contexts [23]. Numerous studies have documented the pervasive influence of platforms like Instagram, Facebook, and TikTok on body image dissatisfaction, especially in younger populations. Adolescents and young adults are particularly vulnerable due to their high engagement with social media, frequent exposure to idealized body images, and the prominence of peer comparison. Studies indicate that women are more likely to experience body dissatisfaction and disordered eating behaviors compared to their male counterparts [24]. A recent systematic review highlights the correlation between social media usage and body image disturbance, particularly among women aged 18–25, with Instagram showing the strongest link due to its image-centric nature [23]. In contrast, older adults show less impact, as suggested by a meta-analysis, which posits that body image concerns in middle-aged adults may be influenced more by real-life social networks than by social media [25,26,27].

Settings also play a role, with Westernized cultures reporting higher rates of body dissatisfaction compared to non-Western countries [24,28,29,30]. Previous research, including the Tripartite Influence Model, established that sociocultural influences from social media, body image concerns, and body shape dissatisfaction are primary determinants of disordered eating behaviors in young women [31]. However, there is limited exploration of how these factors interact specifically among young women in non-Western settings, particularly in Aotearoa, New Zealand, where unique cultural dynamics might influence perceptions of beauty and health.

Social media exposure can shape perceptions of healthy eating and body image, but more research is needed to understand its effects on eating behaviors and dietary intake [23,25]. Given the critical role of both nutrition and psychosocial factors in shaping body image, this study aims to explore the relationship between body image disturbance among young women, diet quality, as measured by Australian Recommended Food Scores (ARFS) [11], and eating behaviors using the Three-Factor Eating Questionnaire [32] (TFEQ-R18). By combining qualitative insights from focus group discussions with quantitative analysis through the Body Image Disturbance Questionnaire [2] (BIDQ), ARFS, and TFEQ-R18, this research seeks to provide a comprehensive understanding of how these factors interact, particularly in the context of social media exposure. The findings will contribute to the development of targeted interventions that address both dietary behaviors and the psychological impacts of media on body image.

## 2. Materials and Methods

### 2.1. Study Design

This study employed a mixed-methods approach to investigate the relationship between diet quality and body image disturbance among young women aged 18–24. The study was conducted in two phases: a qualitative phase involving focus group discussions and a quantitative phase involving the administration of validated questionnaires. This design allowed for a comprehensive exploration of both the psychological and nutritional factors influencing body image in this demographic.

### 2.2. Participants

A total of 19 young women participated in the qualitative phase of the study [33]. These participants were recruited through social media platforms and university bulletin boards. Inclusion criteria for workshops were as follows: identify as female or non-binary, aged 18–24 years, speak fluent English, reside in New Zealand, and have access to an internet-connected device [34]. Participants included young women from various occupational and educational backgrounds, with eight identifying as New Zealand European, eight as Asian (Chinese, Hong Kong Chinese, Korean, and Indian), two as Fijian Indian, and one as European. The median age of workshop participants was 21 years. Most participants were university students (63%) and resided in Auckland [33]. The most used social apps by the cohort were Instagram and TikTok; most participants reported spending an average of 1–2 h per day on their preferred app. When asked to rank their relationship with social media, with ‘5’ being a very positive relationship and ‘1’ indicating an overall negative relationship, the mean across participants was 3.07. Participants provided informed consent and were assured that their contributions would be anonymized. Participants with a history of diagnosed eating disorders or those who indicated concerning disordered eating patterns following preliminary administration of the revised 18-item Three-Factor Eating Questionnaire (TFEQ-r18) were contacted with content trigger warnings to confirm participation in the workshops (*n* = 2) [35].

For the quantitative phase, a separate group of 50 young women, aged 18–24, were recruited through university mailing lists and online advertisements. In addition to the criteria for qualitative phase recruitment, young women were asked to complete screening questions pertaining to exercise and fruit and vegetable intake, as well as social media literacy, concurrent program participation, and medical history (Table 1). Young women who met national guidelines for fruit intake, vegetable intake, and exercise were excluded from participation, as were those participating in alternative healthy lifestyle programs. All participants provided informed consent prior to formal enrolment. As with the qualitative phase, the mean age of the participants in the quantitative phase was 21 years. The majority of participants identified as New Zealand European (60%) and were university students at the time of participation (74%) (Table 2). At the time of recruitment, almost all participants were ‘very’ or ‘extremely’ familiar with social media and engaged with the platforms daily.

### 2.3. Setting and Procedures

The qualitative phase involved semi-structured workshop discussions designed to explore participants’ perceptions of the influence of social media on their body satisfaction and dietary behaviors, as well as how social media could be used to address these aspects of nutrition status [34]. The discussions were conducted via the web-conferencing software Zoom ((Version 5.14.10 (19202)) with the 19 participants in small groups of 2–9, each lasting approximately 60 min [37]. The sessions were audio-recorded and transcribed verbatim for subsequent thematic analysis. A comprehensive overview of the methodological process and insights from thematic analysis can be found in a separate publication and associated protocol paper [33,34]. As aforementioned, qualitative phase participants completed the Three-Factor Eating Questionnaire-R18 (TFEQ-r18), a psychological tool used to assess eating behaviors, prior to study enrolment [35].

The quantitative phase involved the administration of four validated instruments to a separate group of 50 participants: the Body Image Disturbance Questionnaire (BIDQ), the Short Form Food Frequency Questionnaire (SF-FFQ), the TFEQ-r18, and a social influence questionnaire (SIQ) [38,39,40]. All questionnaires rely on self-reports and were administered to participants using Research Electronic Data Capture (REDCap) tools (https://projectredcap.org/, accessed on 19 September 2024) hosted at the University of Auckland [41,42]. Participants were given one week to complete all surveys prior to expiry.

### 2.4. Study Measures

Each questionnaire was administered to assess a different facet of nutrition intake and behavior. The BIDQ was used to determine the extent of body image disturbance experienced by participants [38]. BIDQ responses were coded from 1 to 5. The average of all items (*n* = 7) comprised total BIDQ scores, where higher scores indicate greater body image concerns. Cronbach alpha for the validity of the questionnaire in a population of adult women was 0.89, suggesting instrument reliability and validity in negative body image assessment [38].

The TFEQ-r18 is a tool used to assess disordered eating behaviors. Eating behaviors are evaluated in three realms: cognitive restraint, emotional eating, and uncontrolled eating [35]. Of the 18 items, *n* = 9 assess uncontrolled eating, *n* = 6 assess cognitive restraint (CR), *n* = 9 assess uncontrolled eating (UE), and *n* = 3 assess emotional eating (EE). Survey responses are coded from 1 to 4, with higher scores suggesting concerning disordered eating behaviors. Total scores for each sub-scale were standardized using the following equation: [standardized score = (raw score − minimum possible score)/maximum possible score − minimum possible score × 100]. Cronbach alpha for the TFEQ-r18 has been reported as 0.75 for CR, 0.89 for UE, and 0.87 for EE in a population of young Finnish females [35]. The results of this survey allow for comment on the extent of disordered eating and the investigation of associations between these behaviors and social media use. For the quantitative phase, CR was unable to be reported with validity due to an error in survey dissemination, whereby one question was omitted. Therefore, only UE and EE can be determined reliably for the second phase of the study.

The SS-FFQ and ARFS were used to evaluate dietary intake. The SF-FFQ asks participants to recall food intake over the preceding three months and considers an extensive variety of foods from each of the main food groups [39]. The 70-item Australian Recommended Food Score (ARFS) was then used by the student researcher to determine individual and cohort diet quality, with higher scores reflecting better adherence to recommended dietary guidelines [11]. Most foods are awarded one point for a consumption frequency of >once per week [11]. The ARFS scoring tool was adapted to suit Short Form-FFQ responses. The median intra-class correlation co-efficient for ARFS nutrients has been reported as 0.66 (0.48–0.84), affirming use for the brief assessment of dietary quality [11].

Finally, the SIQ, which includes four items with a 7-point Likert scale, was administered to measure the effect of social influence [40].

In tandem with collated qualitative insights, the surveys aid in the formation of a greater understanding of the dynamic relationship between dietary intake and behavior, body image, and social media use.

### 2.5. Sample Size

For the qualitative phase, there was no set sample size objective. Instead, the research team was focused on balancing sample adequacy with sufficient individual participation. Based on co-design recommendations, the research team aimed to have up to 10 individuals [43]. For the quantitative phase, a recruitment target of *n* = 50 was set based on a systematic review of similar pilot studies, which identified a median total sample size of 30.5 in non-drug trials [44]. The quantitative phase was designed to evaluate the feasibility of research procedures [34]. Due to the nature of a pilot feasibility trial, we did not employ a powered sample size calculation for this study, as the primary aim was not to establish definitive efficacy or achieve statistical power for hypothesis testing, but rather to gather preliminary data on the feasibility and acceptability of intervention components including recruitment strategies, retention rates, and the logistical aspects of conducting a larger trial. The findings from this pilot study will inform the design of a larger-scale study, allowing us to better estimate the necessary sample size for future research focused on the primary outcomes.

### 2.6. Data Analysis

Thematic analysis was conducted on the focus group transcripts using the six-phase approach outlined by Braun and Clarke [34,36]. This method facilitated the identification of key themes related to body image disturbance and dietary behaviors, which were used to contextualize the quantitative findings. Quantitative data were analyzed using SPSS version 26 [45]. Descriptive statistics were calculated for both BIDQ and ARFS. Pearson’s correlation coefficient was employed to examine the relationship between diet quality (ARFS), body image disturbance (BIDQ scores), and disordered eating (TFEQ-r18 scores) among the 50 participants in the quantitative phase. A multiple regression analysis was conducted to examine the relationship between SIQ and BIDQ, TFEQ, and ARFS. A *p*-value of <0.05 was considered statistically significant. Assumptions for multiple regression were tested via normality, a Durbin–Watson test result no less than 1 and no greater than 3, and a standardized residual result between −3.29 and 3.29. A bivariate analysis was conducted to provide Pearson correlation coefficients and their corresponding *p*-values, showing the strength and significance of relationships between the variables of interest.

### 2.7. Ethical Considerations

The research was approved by the Human Participants Ethics Committee at the University of Auckland on 9 June 2022 for three years (UAHPEC24366). Separate informed consent was obtained from participants in both phases. All data were anonymized to ensure confidentiality, and participants were informed of their right to withdraw from the study at any time.

## 3. Results

The qualitative phase of this study aimed to explore how social media influences the body image and eating patterns of young women aged 18–24 in Aotearoa, New Zealand. Through semi-structured interviews, key themes were identified, shedding light on the complex relationship between social media exposure and body image perceptions, as well as eating behaviors. This section presents the qualitative results of the study, which examines how engagement with social media impacts body image and food relationships among young women in Aotearoa, New Zealand. The findings are organized around key themes identified through qualitative analysis, providing insights into the complex links between social media usage, body image perceptions, and eating behaviors.

### 3.1. Engagement with Social Media Platforms and Body Image Perceptions

The study revealed that social media platforms propagate unrealistic beauty ideals through image manipulation and the facilitation of negative social comparisons (Table 3). The predominant themes indicate that constant exposure to these narrow views of beauty can contribute to negative self-perception and body dissatisfaction among young female platform users.

### 3.2. Social Media Influence on Eating Behaviors and Attitudes Towards Diet

The study also assessed social media’s impact on young women’s relationships with food (Table 4). The results highlighted how social media often promotes unrealistic diet and nutrition advice, leading to unhealthy attitudes and behaviors.

### 3.3. Cultural and Contextual Factors Unique to Aotearoa, New Zealand

The discussions on sociocultural contexts revealed that social media’s impact on body image and food relationships is shaped by the unique cultural backdrop of Aotearoa, New Zealand (Table 5). The study critiques the dominance of Western-centric notions of health and emphasizes the need for culturally sensitive interventions.

### 3.4. Diet Quality, Disordered Eating, and Body Image Disturbance

The quantitative phase of this study involved the assessment of diet quality via SF-FFQ and associated ARFS, and body image disturbance via BIDQ administration. Disordered eating behaviors were assessed for both research phases using the TFEQ-r18. All realms of the TFEQ-r18 were assessed for the qualitative phase; however, cognitive restraint cannot be reported for the quantitative phase.

#### 3.4.1. Diet Quality

Adapted ARFS were calculated for the participants to indicate preliminary diet quality. Of the young women (*n* = 50), almost a third (30%) had a diet quality classified as ‘low’, with an additional 24% classified as having a ‘moderately’ healthy diet.

#### 3.4.2. Disordered Eating Behaviors; Cognitive Restraint, Uncontrolled and Emotional Eating Behaviors

The TFEQ-r18 was administered to all participants (*n* = 69). A significant proportion of young women demonstrated moderate or high levels of uncontrolled eating and emotional eating. For uncontrolled eating, 45% of participants displayed moderate levels, and 10.5% exhibited high levels. In terms of emotional eating, 36.5% of participants demonstrated moderate levels, while 15.5% exhibited high levels.

#### 3.4.3. Body Image Disturbance

The majority of young women who completed the pre-intervention BIDQ (*n* = 50) demonstrated mild to moderate body image disturbance (80%: mild to moderate 72%, moderate to severe 8%), indicating “occasional distress and mild functional impairment”. Scores above 20 generally warrant further clinical assessment; only 4 participants (8%) indicated notably distressing body image concerns.

#### 3.4.4. Multiple Linear Regression Analysis

Test associations between ARFS and BIDQ scores were non-significant (R = 0.155, *p* = 0.282). Associations between the ARFS and disordered eating patterns were also non-significant and weak (uncontrolled eating R = 0.33, *p* = 0.823; emotional eating R = 0.002, *p* = 0.991).

Regression analysis indicated a notable relationship between body image disturbance and uncontrolled eating; however, this was not significant (R = 0.259, *p* = 0.069). The relationship between BIDQ scores and emotional eating was weaker than that for uncontrolled eating (R = 0.163, *p* = 0.259).

Regarding social media use, associations were tested between body image disturbance, disordered eating behaviors, and self-reported social media use, including the following: frequency of social media use for contacting others, familiarity with social media, frequency of posting, frequency of engagement with content, and use of social media to source health-related information. A significant relationship was found between body image disturbance and level of social media familiarity (R = 0.326, *p* = 0.02).

The analysis revealed no significant relationships between age and the various measures. Age also did not significantly correlate with the frequency of social media use for contacting, familiarity with social media, or health-seeking behaviors on social media, suggesting that age may not be a critical factor influencing these variables in the context of this study. Associations between social influence, diet quality, and body image disturbance were explored; however, no significant relationships were found. The multiple linear regression analysis has been presented in Table 6 and Table 7. Model summary statistics have been included in the Appendix A.

#### 3.4.5. Bivariate Correlation Analysis

The bivariate correlation analysis revealed several significant associations between the study variables (Table 8). A positive correlation was observed between social media contact frequency and social media familiarity (R = 0.370, *p* = 0.008), social media post frequency (R = 0.417, *p* = 0.003), social media engagement frequency (R = 0.525, *p* < 0.001), and social media health-seeking behaviors (R = 0.411, *p* = 0.003). Similarly, social media familiarity was significantly associated with social media post frequency (R = 0.315, *p* = 0.026), social media engagement frequency (R = 0.497, *p* < 0.001), and social media health-seeking behaviors (R = 0.224, *p* = 0.118).

In terms of body image disturbance (BIDQ), there was a significant positive association with social media familiarity (R = 0.328, *p* = 0.020), indicating that higher social media familiarity correlates with greater body image disturbance. However, no significant correlations were observed between BIDQ and other social media variables, including contact frequency, post frequency, and engagement frequency.

Emotional eating and uncontrolled eating were significantly correlated with each other (R = 0.708, *p* < 0.001), but neither showed significant associations with social media variables. Diet quality as measured by the ARFS did not exhibit any significant relationships with the social media variables or the body image disturbance score.

## 4. Discussion

While previous research has documented an association between social media exposure and negative body image, the impact of this exposure on dietary behaviors, specifically diet quality and disordered eating patterns, remains underexplored. This study addresses these gaps by examining the relationships between social media influence, body image disturbance, eating behaviors and diet quality in young women aged 18–24, a demographic particularly susceptible to social pressures regarding appearance.

The study found that while body image disturbance did not significantly correlate with diet quality, social media familiarity was significantly associated with higher body image disturbance. Additionally, a considerable proportion of participants exhibited disordered eating behaviors, with a notable relationship between uncontrolled eating and body image concerns. However, such behaviors were only weakly associated with social media use. However, the focus group discussions revealed that social media remains a significant context where young women navigate their body image and eating habits, which may contribute to unhealthy comparisons.

A noteworthy proportion of participants indicated low diet quality and mild body image disturbance. It is becoming frequently apparent that extended social media use can have damaging effects on the self-perception of young women [3,4,5,6,8,9,10]. Several sub-themes were identified via focus group discussions which illuminated the potential and harms of social media use for nutrition purposes. Participants frequently experienced negative body image perceptions influenced by unrealistic beauty standards propagated on social media. Many expressed feelings of guilt and pressure to conform to idealized images, which exacerbated disordered eating behaviors.

Social media noise likely results in confusion rather than informed users due to excessive and low-quality information [46]. A probable contributor to this is algorithms which favor popularity, and therefore beauty standards; algorithms do not promote, or push based on quality of evidence or professional credentials [47]. It is left to the user to discern fact from fiction when consuming nutrition information, a task which is difficult without an education in health or nutrition.

The social contagion of disordered eating behaviors may permeate via social media, from one’s inner circle to highly influential encounters online, leading to an overt “social pressure to be thin” [48,49]. Instagram is saturated with weight-loss messaging and harmful dietary advice, which fosters poor body image and nutrition misinformation [37]. For example, social media commonly reinforces one-dimensional diet messaging through body ideals (i.e., “she eats this way, therefore her body is this way” or “she achieves the standard of beauty, therefore her commentary on dietary intake must be correct”). There is little nuanced discussion of genetics, environmental, and other pertinent external influences on nutrition status. The omission of these influences in “Insta norms” such as “fitspiration” can lead to an inflated sense of personal responsibility and, subsequently, a harmed perception of self [10,48].

Despite the obvious pitfalls of platforms like Instagram, this study found limited associations between social media use and diet quality, unlike some studies that suggest social media influences eating behaviors more directly [4,5,6]. No significant relationship was found between social media influence or use and actual diet quality or disordered eating. Conversely, studies by Holland and Tiggemann (2016) and Fardouly et al. (2015) identified social media as a significant factor in body dissatisfaction, leading to unhealthy eating behaviors [24,25]. This may indicate that while social media significantly affects perceptions of body image, its direct impact on eating behaviors may be mediated by other factors such as individual psychological traits or external support systems [24]. The nature of social media content consumed by participants or the possibility that other factors, such as personality traits or environmental influences, mediate the relationship between social media exposure and eating behaviors, is likely [24,50,51,52,53].

Additionally, our findings could reflect the growing awareness of media literacy and body positivity movements, which may buffer the negative impact of social media [54,55,56]. The social cognitive theory suggests that behaviors are influenced by personal, environmental, and behavioral factors, including observational learning from media. This posits that individuals are becoming more adept at critically evaluating the content they consume, which reduces its direct influence on their eating behaviors [57]. As the focus group results demonstrated, some participants were able to critically evaluate the unrealistic standards set by social media and resist conforming to harmful dietary trends. Several participants in this study acknowledged the growing awareness of media manipulation and unrealistic portrayals on social media, which may have allowed some individuals to develop resilience against the harmful effects of these platforms.

A significant body of research, including systematic reviews and meta-analyses, has highlighted how social media exacerbates body image issues and disordered eating behaviors, particularly among young women [10,23]. Studies have demonstrated that increased exposure to idealized body images on social media platforms correlates with body dissatisfaction, restrictive eating, and binge eating behaviors [10]. However, this study’s findings challenge these findings, indicating that the relationship might be context dependent. The Body Image Dissatisfaction Model suggests that the internalization of media ideals is not universal and may depend on individual differences in self-esteem, media literacy, and societal attitudes towards body image [58]. Our findings suggest that social media may no longer universally dictate body dissatisfaction and disordered eating behaviors in this demographic, possibly due to these mediating factors.

Cultural or geographical factors, for example, might attenuate how social media influences body image and eating behaviors [5]. Cultural values, societal norms, and exposure to different types of content can vary greatly between regions. For example, New Zealand culture places a significant emphasis on body diversity, health, and fitness, which could promote more positive attitudes toward body image and less susceptibility to disordered eating behaviors [59]. Conversely, studies conducted in other countries with stronger “thin ideal” cultural norms, such as the US or Western Europe, may find a more pronounced relationship between social media influence and eating behaviors [5]. In countries like New Zealand, where this study took place, a greater emphasis on healthy eating campaigns and critical media consumption may mitigate the negative effects of social media [33,34]. Several participants in this study acknowledged the growing awareness of media manipulation and unrealistic portrayals on social media, which may have allowed some individuals to develop resilience against the harmful effects of these platforms.

One limitation of the study is the relatively small sample size, which may limit the generalizability of the findings. The sample size of 50 participants, while appropriate for a feasibility RCT [34], may have limited the power to detect significant relationships, and a larger sample may yield different results. Conducting multiple regression provided preliminary insights into potential relationships and trends to guide future research. These initial associations help to identify which factors warrant further investigation with larger, more powered samples and can inform the development of more targeted hypotheses and intervention strategies in subsequent studies. Secondly, the cross-sectional design of the study restricts the ability to draw causal conclusions about the relationships between social media exposure, body image disturbance, and eating behaviors. We cannot conclusively determine whether social media influences body image and eating behaviors or if individuals predisposed to these issues are more likely to engage with certain types of social media content. The specific social media platforms and content consumed by participants were not deeply explored, which could account for the differences between our findings and those of other studies. Another potential limitation is the reliance on self-reported data, which can be subject to biases such as social desirability. The dietary intake assessment questionnaire selected for the quantitative phase of the research, the short-form food frequency questionnaire, asks participants to recall intake over a preceding three-month period [39]. Food frequency questionnaires are notoriously prone to recall and underreporting bias, which may preclude the validity of results [60,61].

Future research should explore these relationships longitudinally to better understand the directionality of these associations. It would be valuable to investigate the specific types of social media content consumed (e.g., fitness influencers versus body-positive content) and their differential impacts on eating behaviors and diet quality [62]. Furthermore, exploring the role of cultural and geographical factors in shaping the relationship between social media, body image, and eating behaviors could provide more insights. For example, investigations into potential moderating factors, such as the role of social support or individual psychological resilience, in mediating the impact of social media on both body image and eating behaviors is needed; as well as larger and more diverse samples to provide a broader perspective on how these relationships manifest in different demographic groups.

Finally, for dietitians and healthcare providers, future studies could focus on interventions aimed at increasing media literacy and promoting positive body image to mitigate any potential negative effects of social media [33,34,63]. The fostering of these skills would likely be beneficial to clinical work with young people, whereby the provision of social media advice alongside general nutrition guidance could act to build confidence in content quality discernment and protection against harmful body ideals.

It is vital to gain a richer understanding of how social media algorithms, and their influential users, impact the nutrition status of young women. Beyond comprehension, it is important to investigate how best to utilize social platforms to improve the body image and dietary intake of young people. This study may inform future research attempting to achieve such objectives. The Daily Health Coach feasibility trial is a pilot RCT evaluating the impact of a 3-month healthy lifestyles program on Instagram for young women in Aotearoa, NZ. We call for similar research to be undertaken with distinct population groups in order to collectively map how best to navigate social apps from the practitioner and research perspective.

## 5. Conclusions

This study investigates the relationship between body image, eating behaviors, and diet quality in young women aged 18–24, with a specific focus on the impact of social media. The findings suggest that social media exposure is significantly associated with body image disturbance in this demographic. However, while social media plays a substantial role in shaping body image perceptions, its direct impact on eating behaviors, such as uncontrolled and emotional eating, and diet quality, is less pronounced. The study reveals only weak correlations between social media influence and actual eating patterns or diet quality, suggesting that while social media fosters body image concerns, other factors such as media literacy, cultural attitudes, and individual personality traits may moderate its effect on dietary behaviors. These results highlight the complexity of the relationship between body image, eating behaviors, and diet quality, indicating that interventions should address the psychological drivers behind these concerns, rather than solely focusing on reducing social media exposure.

## Figures and Tables

**Table 1 nutrients-16-03517-t001:** Inclusion and exclusion criteria for qualitative and quantitative study phases.

Inclusion Criteria	Exclusion Criteria
Aged 18 to 24 years.	Individuals unable to given informed consent due to diminishing comprehension or understanding and/or those with a disability (e.g., sight or hearing impairment) that precludes participation.
Identifies as female or non-binary.	Self-reported meeting national recommendations for fruit and vegetable intake (based on age/sex recommendations) ^a^ and self-reported meeting physical activity recommendations ^b^ [36].
Social media literate according to study-specific criteria (outlined below).	Non-English speaking.
Available for intervention.	Currently participating in an alternative healthy lifestyle program.
Access to a computer or tablet or smartphone with e-mail and internet facilities.	History or major medical problems ^c^ that had not been granted GP approval to participate ^d^ and/or diagnosed with an active eating disorder.

^a^ Five servings of vegetables and two servings of fruit per day. ^b^ At least 2 ½ h of moderate (30 min/day) or 1 ¼ h of vigorous physical activity spread throughout the week. ^c^ Including heart disease or diabetes that requires insulin injections. ^d^ Those answering ‘yes’ to any of the conditions in the pre-medial exercise screener require GP approval to participate.

**Table 2 nutrients-16-03517-t002:** The baseline characteristics of the participants in the quantitative phase.

Variable	Total (*n* = 50) Mean (SD) or % (*n*)
Age	21.34 (0.50508)
Ethnicity:	
New Zealand European	62% (31)
Chinese	18% (9)
Indian	4% (2)
Korean	4% (2)
Other (such as Japanese, Indonesian, Taiwanese)	12% (6)
Employment Status:	
Currently studying/student	74% (37)
Employed, working 40 or more hours per week	16% (8)
Employed, working less than 40 h per week	10% (5)
Social Media Frequency of Use:	
Never	0
Every couple of weeks	2% (1)
Multiple times a day	6% (3)
Daily	30% (15)
Multiple times a day	62% (31)
Social Media Familiarity:	
Not familiar at all	0
Slightly familiar	0
Moderately familiar	4% (2)
Very familiar	18% (9)
Extremely familiar	78% (39)
Social Media Engagement:	
A few times a year	2% (1)
A few times a month	4% (2)
Weekly	6% (3)
Multiple times a week	10% (5)
Daily	42% (21)
Multiple times a day	36% (18)
Social Media Health Seeking Behaviors	
Never	2% (1)
Very occasionally	26% (13)
Sometimes	28% (14)
Often	36% (18)
All of the time	8% (4)

SD: Standard deviation.

**Table 3 nutrients-16-03517-t003:** Summary of themes related to social media and body image perceptions.

Theme	Description	Quotes
Unrealistic Beauty Standards	Social media promotes unattainable beauty ideals through image manipulation.	“I’ve seen a lot of those like…people showing what they look like when they’ve photoshopped themselves.” (Participant 7)
Social Comparison	Social media facilitates comparisons to idealized standards, intensifying appearance fixation.	“The comparison part of social media is really, really strong.” (Participant 2)
Pressure to Conform	There is pressure to conform to beauty standards and dietary practices seen on social media.	“You feel like you have to follow the latest diet trend to fit in.” (Participant 15)
Food Guilt and Disordered Eating	Diet culture on social media leads to food guilt and disordered eating behaviors.	“Every time I eat something ‘bad’, I feel so guilty because of what I see online.” (Participant 5)
Positive Influence	Social media can also provide positive learning experiences and body acceptance.	“I have learnt from social media posts on how I can improve how I see myself.” (Participant 5)

**Table 4 nutrients-16-03517-t004:** Social media’s influence on eating behaviors.

Sub-Theme	Description	Quotes
Need for Social Media in Dieting	Young women turn to social media for dieting advice, often encountering unrealistic diets.	“I’ve gone to different websites to find new diets … but it goes well for the first couple of weeks and then I get tired.” (Participant 7)
Nutrition and Social Life Influence	Social media often neglects the social and cultural aspects of nutrition.	“Nutrition is so much more than just food; it’s about your social life.” (Participant 1)
Nutrition Influence in Social Media	Social media content focused on diet and fitness often promotes unhealthy behaviors.	“Posting what I eat in a day…it’s very restrictive and sets unrealistic standards.” (Participant 12)
Social Media Influence on Dieting	Social media is a popular source for recipe inspiration and nutritional guidance, though often lacking credible expertise.	“There’s heaps of Instagram influencers … providing nutritional advice.” (Participant 2)
Food Eating Influence	Social media shapes young women’s eating habits and attitudes, often promoting disordered behaviors.	“I go on social media and see the messages about eating healthy … it starts a whole negative cycle.” (Participant 5)

**Table 5 nutrients-16-03517-t005:** Cultural and contextual factors influencing social media impact.

Theme	Description	Quotes
Communities and Culture on Recipes	The influence of the community and cultural backgrounds on food preferences and social media engagement.	“Westernized recipes … are not relatable or engaging for a young audience from different cultures.” (Participant 3)
Cultural Context Effect on Dieting	Cultural upbringings significantly shape attitudes and behaviors around food and dieting.	“Food is so much more than just what we eat; it’s the context in which we eat it.” (Participant 9)
Cultural Appropriation	The appropriation of ethnic food by white social media influencers contributes to cultural erasure.	“It annoys me when this food is taken from a different culture and a white person is cooking it.” (Participant 7)
New Zealand Culture on Body Image	The narrow definition of health in New Zealand often excludes minority groups, contributing to poor body image among young women.	“Health is often visualized as abled, white, slim, which is not inclusive.” (Participant 1)

**Table 6 nutrients-16-03517-t006:** Multiple regression analysis for Body Image Disturbance (BIDQ), social influence questionnaire (SIQ), Australian Recommended Food Score (ARFS).

					Un-Std. Coeff.			Std. Coeff.
Dependent Variable	Independent Variable	Significance (*p*-Value)	Effect Size ^a^ (R-Value)	B	B (Constant)	Std. Error	Std. Error (Constant)	Beta
Body Image Disturbance (BIDQ)	Social Media Contact Freq.	0.600	0.076	−0.464	16.458	0.880	4.026	−0.076
	Social Media Familiarity	0.020 *	0.328	2.693	1.595	1.118	5.330	0.328
	Social Media Post Freq.	0.443	0.111	0.433	13.493	0.560	1.278	0.111
	Social Media Engagement Freq.	0.423	0.116	0.423	12.273	0.523	2.655	0.116
	Social Media Health Seeking	0.123	0.221	0.987	11.202	0.628	2.098	0.221
	ARFS	0.282	0.155	0.110	11.455	0.102	2.742	0.155
	Uncontrolled Eating	0.069	0.259	0.063	11.448	0.034	1.674	0.259
	Emotional Eating	0.259	0.163	0.026	13.133	0.023	1.235	0.163
Social Influence (SIQ)	ARFS	0.518	0.094	0.316	99.510	0.485	13.082	0.094
	BIDQ	0.958	0.008	0.036	107.299	0.683	10.239	0.008
	Uncontrolled Eating	0.842	0.029	−0.033	109.354	0.166	8.204	−0.029
	Emotional Eating	0.630	0.070	−0.053	110.311	0.109	5.913	−0.070
Diet Quality (ARFS)	Uncontrolled Eating	0.823	0.033	−0.011	26.832	0.049	2.433	−0.033
	Emotional Eating	0.991	0.002	0.000	26.303	0.032	1.758	0.002

^a^ Pearson correlation coefficient. * Correlation is significant at the 0.05 level (2-tailed).

**Table 7 nutrients-16-03517-t007:** Multiple regression analysis for relationship assessment between social media use for two dimensions of Three-Factor Eating Questionnaire (TFEQ: Uncontrolled Eating and Emotional Eating).

				Un-Std. Coeff.				Std. Coeff.
Dependent Variable	Independent Variable	Significance (*p*-Value)	Effect Size ^a^ (R-Value)	B	B (Constant)	Std. Error	Std. Error (Constant)	Beta
Uncontrolled eating (TFEQ)	Social Media Contact Freq.	0.802	0.036	0.914	42.018	3.627	16.590	0.036
	Social Media Familiarity	0.263	0.161	5.434	20.392	4.802	22.901	0.161
	Social Media Post Freq.	0.380	0.127	2.037	42.074	3.519	8.024	0.127
	Social Media Engagement Freq.	0.855	0.026	0.397	44.186	2.164	10.987	0.026
	Social Media Health Seeking	0.459	0.107	1.964	39.862	2.633	8.795	0.107
Emotional eating (TFEQ)	Social Media Contact Freq.	0.728	0.050	−1.924	55.810	5.505	25.177	−0.050
	Social Media Familiarity	0.732	0.050	2.545	35.047	7.380	35.192	0.050
	Social Media Post Freq.	0.834	0.030	0.741	45.630	3.519	8.024	0.030
	Social Media Engagement Freq.	0.232	0.172	−3.920	66.476	3.238	16.441	−0.172
	Social Media Health Seeking	0.702	0.055	−1.546	52.058	4.014	13.411	−0.055

^a^ Pearson correlation coefficient.

**Table 8 nutrients-16-03517-t008:** Bivariate correlations between social media use, body image disturbance, eating behaviors, and diet quality.

	SIQ ^a^	BIDQ ^b^	TFEQ ^c^ (UE ^d^)	TFEQ (EE ^e^)	ARFS ^f^	SM ^g^ Contact Frequency ^h^	SM Familiarity	SM Post Frequency	SM Engagement Frequency	SM Health Seeking Behavior
SIQ	1	0.008 ^i^ (*p* = 0.958)	−0.029 (*p* = 0.630)	−0.070 (*p* = 0.630)	0.094 (*p* = 0.518)	0.255 (*p* = 0.074)	0.173 (*p* = 0.229)	0.245 (*p* = 0.086)	0.095 (*p* = 0.513)	0.151 (*p* = 0.295)
BIDQ	0.008 (*p* = 0.958)	1	0.259 (*p* = 0.069)	0.163 (*p* = 0.259)	0.155 (*p* = 0.282)	−0.076 (*p* = 0.600)	0.328 * (*p* = 0.020)	0.111 (*p* = 0.443)	0.116 (*p* = 0.423)	0.221 (*p* = 0.123)
TFEQ (UE)	−0.029 (*p* = 0.842)	0.259 (*p* = 0.069)	1	0.708 ** (*p* < 0.001)	−0.033 (*p* = 0.823)	0.036 (*p* = 0.802)	0.161 (*p* = 0.263)	0.127 (*p* = 0.380)	0.026 (*p* 0.855)	0.107 (*p* = 0.459)
TFEQ (EE)	−0.070 (*p* = 0.630)	0.163 (*p* = 0.259)	0.708 ** (*p* < 0.001)	1	0.002 (*p* = 0.991)	−0.050 (*p* = 0.728)	0.050 (*p* = 0.732)	0.030 (*p* = 0.834)	−0.172 (*p* = 0.232)	−0.055 (*p* = 0.702)
ARFS	0.094 (*p* = 0.518)	0.155 (*p* = 0.282)	−0.033 (*p* = 0.823)	0.002 (*p* = 0.991)	1	0.113 (*p* = 0.436)	0.275 (*p* = 0.053)	0.015 (*p* = 0.917)	−0.082 (*p* = 0.569)	0.017 (*p* = 0.909)
SM Contact Frequency	0.255 (*p* = 0.074)	−0.076 (*p* = 0.600)	0.036 (*p* = 0.802)	−0.050 (*p* = 0.728)	0.113 (*p* = 0.436)	1	0.370 ** (*p* = 0.008)	0.417 ** (*p* = 0.003)	0.525 ** (*p* < 0.001)	0.411 ** (*p* = 0.003)
SM Familiarity	0.173 (*p* = 0.229)	0.328 * (*p* = 0.020)	0.161 (*p* = 0.263)	0.050 (*p* = 0.732)	0.275 (*p* = 0.053)	0.370 ** (*p* = 0.008)	1	0.315 * (*p* = 0.026)	0.497 ** (*p* < 0.001)	0.224 (*p* = 0.118)
SM Post Frequency	0.245 (*p* = 0.086)	0.111 (*p* = 0.443)	0.127 (*p* = 0.380)	0.030 (*p* = 0.834)	0.015 (*p* = 0.917)	0.417 ** (*p* = 0.003)	0.315 * (*p* = 0.026)	1	0.280 * (*p* = 0.049)	0.438 ** (*p* = 0.002)
SM Engagement Frequency	0.095 (*p* = 0.513)	0.116 (*p* = 0.423)	0.026 (*p* = 0.855)	−0.172 (*p* = 0.232)	−0.082 (*p* = 0.569)	0.525 ** (*p* < 0.001)	0.497 ** (*p* < 0.001)	0.280 * (*p* = 0.049)	1	0.419 ** (*p* = 0.002)
SM Health Seeking Behavior	0.151 (*p* = 0.295)	0.221 (*p* = 0.123)	0.107 (*p* = 0.459)	−0.055 (*p* = 0.702)	0.017 (*p* = 0.909)	0.411 ** (*p* = 0.003)	0.224 (*p* = 0.118)	0.438 ** (*p* = 0.001)	0.419 ** (*p* = 0.002)	1

^a^ Social Influence Questionnaire. ^b^ Body Image Disturbance Questionnaire. ^c^ Three-Factor Eating Questionnaire. ^d^ Uncontrolled Eating, ^e^ Emotional Eating. ^f^ Australian Recommended Food Score. ^g^ Social Media. ^h^ Use of social media to contact others and share information. ^i^ Pearson correlation coefficient. * Correlation is significant at the 0.05 level (2-tailed). ** Correlation is significant at the 0.01 level (2-tailed).

## Data Availability

The data presented in this study are available on request from the corresponding author. The data are not publicly available due to privacy or ethical restrictions.

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
