# Peer review of "Associations Between Body Image, Eating Behaviors, and Diet Quality Among Young Women in New Zealand: The Role of Social Media"

_nutrients, 2024, doi:10.3390/nu16203517_

Round 1

Reviewer 1 Report

Comments and Suggestions for Authors

Dear Authors,

Thank you for submitting your manuscript. Please find my comments below.

In the Introduction section, it is unclear what novel contribution this study makes to the existing literature. Previous research, such as the Tripartite Influence Model developed by Thompson et al. (1999), has established that sociocultural influences from social media, body image concerns, and body shape dissatisfaction are primary determinants of disordered eating behaviors in young women. Therefore, the study’s novelty should be articulated more clearly for an international audience. Additionally, the study's aims should be explicitly stated at the end of the Introduction section, and subsequent data analysis should align with these aims. Furthermore, the term "diet quality" is not clearly defined, nor is its relationship to other variables under investigation adequately explained.

The Methods section is disorganized and should be restructured in accordance with the STROBE guidelines, specifically: 1. Study design. 2. Participants. 3. Settings and Procedure. 4. Study Measures. 5. Sample size. 6. Data Analysis. The study measures (e.g., questionnaires, scales) should be described in greater detail. This includes providing comprehensive information on subscales, procedures for calculating final scores, and Cronbach's alpha values to demonstrate internal consistency within this study.

The Results section is poorly structured. The findings should be presented based on statistical analysis, with correlation and linear regression analyses separated into distinct tables. Additionally, it is unclear why certain variables, such as uncontrolled eating and emotional eating, are simultaneously used as both dependent and independent variables (as shown in Table 2). The results are missing standardized and unstandardized regression coefficients, as well as model summary statistics. Moreover, justifying the sample size based on a "systematic review of pilot studies that identified a median total sample size of 30.5 in non-drug trials" (lines 117-118) is not appropriate. Given that most of your findings are not statistically significant, it suggests the study may be underpowered and susceptible to multiple Type II errors due to an inadequate sample size.

Additionally, the tables are not numbered correctly, as the sequence goes from Tables 1, 2, 3, and then repeats with Table 2.

Lastly, the referencing style and reference list do not adhere to the journal’s formatting guidelines.

Author Response

Thank you very much for taking the time to review this manuscript. Please find attached the detailed responses and the corresponding revisions/corrections highlighted/in track changes in the re-submitted files

Reviewer 2 Report

Comments and Suggestions for Authors

This is a good and interesting study. However, it has some shortcomings that should be addressed:

- In relation to the theoretical framework of the manuscript, I consider that the presentation of the key concepts in the research should be improved. For example, what do we mean by Body Image? What do we mean by quality diet? Definitions from leading authors in the field, such as Cash, Thompson, Tiggemann, Grogan, etc. need to be added to clarify these concepts.

- Throughout the Theoretical Framework, statements are made without a clear scientific basis. For example, in lines 67-70 it is stated that ‘Despite these findings, there remains a gap in longitudinal studies that explore how social media exposure affects body image over time and whether these effects differ in non-Western settings or among older populations’. Regarding ‘non-Western settings’ one should cite works such as:

https://doi.org/10.1016/j.bodyim.2012.06.002

https://doi.org/10.1016/j.bodyim.2009.10.003

https://doi.org/10.1002/erv.678

Regarding ‘among older populations’ you should cite works such as:

https://doi.org/10.3389/fpsyg.2019.02823

https://doi.org/10.1017/S0144686X09008721

https://doi.org/10.15448/1980-6108.2013.4.15357

- It is cited in the study objectives that longitudinal assessment is a gap in this area of study, but I do not understand the connection of that gap to this study. Please expand on that explanation so that I can understand how this study contributes to that gap.

- Regarding the sampling done. It is not clear what the selection criteria for participants are and what the target population is.

- Why does the study not assess men? It is stated that body image is a problem among women, but it should be specified whether this is exclusively the case and thus justify why only women are included in the study.

- Regarding the assessment instruments, they should be presented in the Method section with their psychometric properties and their validity, reliability and the reasons for their selection in the study should be described. 

Author Response

Thank you very much for taking the time to review this manuscript. Please find the detailed responses attached and the corresponding revisions/corrections highlighted/in track changes in the re-submitted files.

Round 2

Reviewer 1 Report

Comments and Suggestions for Authors

Dear Authors,

Thank you for the improved version of your manuscript.

I have several suggestions. Please add a correlation matrix between the study variables. Since many of the effects in the linear regression are not statistically significant, simple bivariate correlations would be helpful to understand the nature of associations. Also, independent variables can mask each other's effects in a multivariable model. Bivariate correlations can reveal individual associations that may be hidden in the full model.

Additionally, I recommend avoiding causal language. For instance, you might change the word "impact" in the study title to "associations." Please review the text carefully and revise it to eliminate any causal language.

Author Response

Reviewer Comment 1: Please add a correlation matrix between the study variables. Since many of the effects in the linear regression are not statistically significant, simple bivariate correlations would be helpful to understand the nature of associations. Also, independent variables can mask each other's effects in a multivariable model. Bivariate correlations can reveal individual associations that may be hidden in the full model.

Response: We appreciate the reviewer’s suggestion to include a correlation matrix to provide a clearer understanding of the relationships between the study variables. We have now added a bivariate correlation matrix to the manuscript (Table 7). This table presents Pearson correlation coefficients for all key variables, including social media usage, body image disturbance, eating behaviors, and diet quality. This analysis helps to clarify individual associations that may have been obscured in the multivariable regression models. We have expanded the results section to include an interpretation of the bivariate correlations.

Reviewer Comment 2: I recommend avoiding causal language. For instance, you might change the word "impact" in the study title to "associations." Please review the text carefully and revise it to eliminate any causal language.

Response: Thank you for this valuable feedback. We have carefully reviewed the entire manuscript and made revisions to eliminate causal language. Specifically, we have replaced the word "impact" with "associations" in the title and throughout the text, ensuring that the language reflects the observational nature of the study. These changes align the manuscript with the appropriate interpretation of our findings based on the cross-sectional design.

All changes and additions have been highlighted in green in the revised manuscript.